# Psychometric Properties of the Action Research Arm Test (ARAT) Scale in Post-Stroke Patients—Spanish Population

**DOI:** 10.3390/ijerph192214918

**Published:** 2022-11-13

**Authors:** Jessica Fernández-Solana, Rocío Pardo-Hernández, Jerónimo J. González-Bernal, Esteban Sánchez-González, Josefa González-Santos, Raúl Soto-Cámara, Mirian Santamaría-Pelaez

**Affiliations:** 1Department of Health Sciences, University of Burgos, 09001 Burgos, Spain; 2Departement of Medicine, University of Vic, 08500 Barcelona, Spain

**Keywords:** action research arm test, function, upper limb, ACV

## Abstract

The validation of measuring instruments in the field of health is a requirement before they can be used safely and reliably. The action research arm test (ARAT) tool is an instrument validated in numerous countries and languages and for different populations, and its use is widespread. The objective of this research was to determine the psychometric properties of ARAT for a sample composed of post-stroke patients. To achieve this, a psychometric analysis was performed, where internal consistency tests were carried out using Cronbach’s alpha, correlations between items and item-total and half-level tests to verify their reliability. Regarding validity, criteria validity tests were performed, taking the motor function dimension of the Fugl–Meyer scale as gold standard, and convergent validity tests were performed by correlation with the FIM–FAM, ECVI-38 and Lawton and Brody scales. The results showed very good internal consistency as well as good criterion and convergent validity. In conclusion, the ARAT can be considered a valid and reliable instrument for the evaluation of upper limb function in post-stroke patients.

## 1. Introduction

Stroke is the most common neurological disease [1] and represents one of the leading causes of death worldwide [2], although rates have declined in recent decades thanks to the development of medical technology [3]. It is also one of the leading causes of long-term disability, with 13.7 million new cases worldwide [2,3,4]. The estimated incidence ranges from between 76 and 119 per 100,000 people per year; but despite medical advances, stroke causes disability in at least 50% of patients who suffer from it [4]. This is a multifactorial disability, and approximately 85% of survivors experience an affectation of the upper and/or lower limbs; these impose limitations in daily life in up to 60% of cases [3,5], which can lead to the disability being chronic [1]. Some of the most common neurological deficits or deficiencies include hemiparesis, cognitive deficits, alterations in muscle tone, strength, sensitivity or visual–spatial perception disorders [6].

Hemiparesis is one of the main deficiencies caused by a stroke that can seriously affect the activities of daily living (ADL) and quality of life (QoL) of the person [7]. Almost 80% of people who suffer from mild paresis will achieve complete upper limb function; however, in the severe cases, only 20% will reach their full function [4]. Likewise, approximately 50–60% of patients will continue to experience some degree of motor impairment of gross motor function. Recovery of gross motor function is essential to restore functionality [6]. 

From a clinical point of view, the lack of early recovery from motor deficits significantly reduces the person’s ability to participate in all activities [7]. The recovery of deficits derived from stroke depends mainly on rehabilitation; however, patients have individual qualities, so it is essential to apply personalized rehabilitation programs based on the levels of motor impairment [8]. Therefore, carrying out a functionality assessment is a very important role before designing the treatment. 

There are several scales that evaluate the functions of stroke patients, and their analysis is essential to establish an adequate evaluation of patients and to be able to compare the results of treatment [6]. Some are included in this paper, such as the Brunnstrom approach, the Fugl–Meyer assessment scale (FMA), the motor assessment scale (MAS), the box and blocks test (BBT), the action research arm test (ARAT) and the functional independence measure (FIM) or functional assessment measure (FAM), among others. The FMA assessment has emerged as the most widely used scale due to its strong psychometric evidence [6,9,10]. To this, it is important to add the measurement of QoL, since it is closely related to health at the physical and functional level; for this, the specific QoL scale for stroke (ECVI-38) stands out [10,11].

According to the clinical assessment of upper limb In neurorehabilitation (CAULIN), which provides evidence-based recommendations for the assessment of upper limbs after stroke, the Fugl–Meyer assessment upper extremity (FMA-EU) and ARAT scales are basic evaluations to measure the functionality of the upper limb in patients suffering from stroke [12]; this is in accordance with the consensus of the Roundtable on Recovery and Rehabilitation (SRRR) [13]. Currently, and based on the systematic review of Prange-Lasonder et al. [12], the use of these two scales is considered whenever possible to cover all aspects of the person’s functioning. Likewise, although they measure different constructs of the framework of the International Classification of Functioning (IPC), there are strong correlations between them [14,15].

In this sense, the ARAT is a standardized observational performance measure that is widely used to assess the functionality of the upper limb after a stroke. It is a scale with satisfactory psychometric properties, including concurrent/convergent validity, reliability and responsiveness [16,17,18]. In terms of psychometric properties, therefore, it has shown that it measures what it intends to measure, which is the limitation of the activity of the upper limbs, and is able to detect changes in the performance of the upper limbs [18].

The European evidence-based recommendations for clinical assessment of upper limb in neurorehabilitation (CAULIN), through a data synthesis extracted from several systematic reviews and guidelines for clinical practice and expert consensus, established that there is a need for an agreed set of reliable, valid and accessible assessment tools to evaluate rehabilitation outcomes, so that relevant scientific evidence can be generated about the effectiveness of different approaches that can be developed during rehabilitation [12]. Among the recommendations is a toolkit divided into three parts: basic tools, an extended toolkit and a recommended toolkit. As basic tools for clinical practice and research, FMA-EU and ARAT are recommended; within the extended set recommended for clinical practice and/or research are the schematic measures, box and block test (BBT), Chedoke Arm Hand Activity Inventory (CAHAI), Wolf motor function test (WMFT), nine hole peg test (NHPT) and ABILHAND; finally, tools recommended for research or specific occasions include the motor index (MI); Chedoke–McMaster stroke assessment (CMSA), stroke rehabilitation assessment movement (STREAM), Frenchay arm test (FAT), motor assessment scale (MAS) and body movement sensors. The CAULIN recommendations provide a clear, simple and evidence-based three-level framework for upper extremity assessment in neurological rehabilitation, of which the ARAT tool is at the first level, making it a tool that is considered basic. The widespread and sustained use of these tools will improve the quality of clinical practice and facilitate the research and meta-analysis necessary for the advancement of neurorehabilitation [12].

The ARAT measure has been validated in several countries [17,19,20,21,22,23] and translated into Spanish; however, it has been adapted only to the Hispanic American population [17,22]. However, some studies conducted in Spain [24,25] have used the version of the scale translated into Spanish, but without validation for this population.

This study corresponds to a study of the reliability and validity of the ARAT instrument in the Spanish population, due to the fact that it is an instrument that allows the assessment of the impairment and functional recovery of the upper limb affected by a stroke [17] and that it is not currently validated. 

Therefore, although it is a widely used evaluation instrument, and despite all previous studies, the ARAT instrument for the evaluation of the functionality of the upper limb has not been validated for post-stroke patients in the Spanish population; this being the main objective that this study pursues.

## 2. Materials and Methods

### 2.1. Participants

This cross-sectional study was conducted at the Neurology and Rehabilitation Services of the University Hospital of Burgos and the Reina Sofia Hospital of Cordoba (Spain). The sample consisted of 83 people who had previously suffered a stroke and were in the acute phase of their recovery, i.e., in the first six months after the episode. The study population was recruited at discharge from the Neurology Service and Stroke Unit of both hospitals between 2019 and 2022 by consecutive sampling, followed by data collection and processing. 

### 2.2. Instruments

A two-part information collection dossier was used. The first consisted of general and socio-demographic data (age, biological gender, education level, marital status, children, family history of stroke, previous stroke, type of stroke, etiology of stroke and affected cerebral hemisphere). The second part included the ARAT, FMA-EU, FIM–FAM, Lawton and Brody and ECVI-38 tools.

ARAT. A measure that evaluates the person’s ability to use their upper limb in the handling of objects through grip, pressure and gross motor movements, which are essential for performing ADLs. In this sense, the inability to perform the test items is proposed as a valid indicator of limitation of the upper limb for activity [26]. The test contains 19 items, in which the participant is asked to manipulate and move objects, starting with the upper limb with the least affectation. It is divided into 4 subscales (grip, take, clamp and gross motor movement). It is scored on an ordinary scale from 0 to 3 (each upper limb is scored separately), with 0 representing non-performance of the test and 3 representing normal execution; possible total scores range from 0 to 57 [18].

It is an appropriate scale to use for people who have suffered a stroke, in addition to being considered a valid and sensitive instrument for measuring the functional limitation of the upper limb in the first weeks after a stroke; therefore, it is an appropriate measure to use in acute post-stroke patients [18,27,28]. 

FMA-EU. A measure used for the evaluation of the functionality of the upper limbs. It contains 33 items that are valued from 0 to 2, with 0 representing non-realization and 2 representing complete realization. It is divided into 5 domains (motor, sensory, balance, range of motion and pain). The score is calculated on 66 points, with partial scores of 36 points for the upper arm and 30 points for wrist and hand. It is translated into Spanish and validated in the Spanish population, and its reliability and validity are demonstrated [29,30,31].

FIM–FAM. A method used to determine the degree of independence in ADLs. The FIM is a global measure of disability that contains 18 items that can be used independently or with 12 additional items belonging to the FAM. It has a scoring system from 1 to 7, with 1 representing total dependence and 7 representing total independence. It scores with a total of 210 and a minimum of 30 points [32,33,34]. 

Lawton and Brody. A method that evaluates the ability to perform the instrumental activities of daily life. It consists of 8 items that are valued at 0 or 1, with 0 representing the non-realization and 1 representing independence [35,36,37].

ECVI-38. A specific questionnaire that measures QoL in patients who have suffered a stroke. It differs from other questionnaires because it includes items that assess vision, cognition and communication, and can be used in those patients who have an ischemic or hemorrhagic stroke and motor aphasia. The Spanish version has demonstrated good validity, reliability and viability. It includes 38 items grouped into 8 domains (physical state, communication, cognition, emotions, feelings, ADLs, common activities of daily living and socio-family functions; with two additional questions about sexual function and work activity). It is valued from 1 to 5, with 1 representing no difficulty in performing the tasks proposed and 5 representing extreme difficulty. This questionnaire is the first instrument developed in the Spanish language to measure QoL specifically in stroke patients [38,39].

### 2.3. Procedure

After the signing of a document of collaboration and confidentiality agreement with the participating centers, the necessary data for this research were taken. The Ethics Committee of the University of Burgos, University Hospital of Burgos and Reina Sofía Hospital of Córdoba positively valued the research plan in the IR Approval Committee HUBU 2134/2019. In each of the participating centers, the person designated for this purpose performed the data collection. The data, which were obtained through the participating centers, were sent to the research team after an anonymization process; from this moment on, they were always treated anonymously and together.

A randomized, controlled and blinded sampling was performed, where recruitment, data collection and analysis of the results were carried out by different members of the research team; no sample calculation was made. In terms of inclusion criteria, participants were considered if they were over 18 years of age with a diagnosis of residual hemiparesis due to stroke, and if the movement of their affected upper extremities was classified between stages II and IV of the Brunnstrom Scale [40]. Criteria for exclusion included the presence of aphasia and cognitive impairment greater than 28 in MOCA [41]. All participants signed an informed consent form before starting.

Once the data were obtained, a matrix was created for evaluation using the statistical program Software IBM SPSS (Statistical Package for the Social Sciences) in its version 25.

### 2.4. Data Analysis

A psychometric analysis of the ARAT scale was performed, for which both a reliability and a validity analysis were carried out.

Firstly, for the reliability analysis, the internal consistency was checked using Cronbach’s alpha, correlations between the items, item-total correlation and half-and-half test. 

For the analysis of validity, the validity of the criterion was checked by comparison with the motor function dimension of the FMA scale, which is considered as gold standard [12]; the convergent validity was checked by correlation with FIMFAM, ECVI-38 and Lawton and Brody scale.

## 3. Results

In the sample of 83 people who had suffered a previous stroke and had an average age of 61.81 years (SD ± 11.540), 44.4% were men and 55.6% were women. Among them, 64.7% had basic education; 57.1% were married or in a couple, followed by those who were separated or divorced (20%), single (14.3%) and widowers (8.6%). Most (88.6%) of the participants had children.

Most participants had no family history of stroke (74.2%), and most had not had a previous stroke (87. 1%). The types of stroke were, from the highest occurrence rate to the lowest, ischemic (87.1%), hemorrhagic (9.7%) and AIT (3.2%). As for their etiology, most had hemorrhagic causes (35.5%), followed by atherothrombotic causes (29%), cardioembolic causes (9.7%) and small vessel disease (9.7%), while the rest (16.1%) had rare or indeterminate causes. Right-hemisphere involvement occurred in 55.2%, while 44.8% cases had left hemisphere involvement. 

### 3.1. Reliability Analysis

#### Internal Consistency

Cronbach’s alpha

Reliability was analyzed using Cronbach’s alpha to check the internal consistency of the instrument; in addition, the item-total correlation was verified; the correlation squared (variance explained) with the items on the scale; and the value of Cronbach’s alpha was determined if each item was deleted. 

The Cronbach’s alpha value obtained was α = 0.96. In addition, Cronbach’s alpha with each of the suppressed elements ranged from 0.958 and 0.960, and the total correlation of corrected elements was greater than 0.42 in all cases.

Correlations between the items that constitute the scale and the item-total correlation

In all cases, high correlations of each of the items that constituted the ARAT appeared with each other and with the total score of the tool. For the right upper limb, the correlation coefficients ranged from 0.771 to 0.941 (*p* < 0.001), and for the left upper limb, the coefficients were between 0.708 and 0.859 (*p* < 0.001).

Half-and-half

The reliability of the ARAT instrument was also examined using the half-and-half test, as shown in Table 1, with a Spearman–Brown value (equal length) indicating very good reliability.

### 3.2. Validity Analysis

#### 3.2.1. Criterion Validity

The total score of the ARAT and its subscales correlated positively with the score of the motor function dimension of the FMA-UE scale (chosen as the gold standard for comparison), with correlation coefficients between 0.369 and .596 (*p* ≤ 0.001); and with the total score of the FMA-EU scale (correlation coefficients between 0.343 and 0.609 (*p* ≤ 0.001)), so that the higher the ARAT scores, the higher the scores in the physical function dimension of FMA-UE and the higher its total score (Table 2).

#### 3.2.2. Convergent Validity

Table 3 shows the correlation of the total score of the ARAT and its subscales with the FINSTRUMENTS IM-FAM, the Lawton and Brody scale and ECVI-38, respectively.

The correlations with the FIM–FAM and Lawton and Brody scales were significant, showing that the worse the functionality of the upper limb, the worse the performance in activities of daily living; there were exceptions in the cases of clamp (right hand) with FIM.FAM, clamp (left hand) with Lawton and Brody and the total score of the ARAT (left hand) with Lawton and Brody, although in the latter case, the same trend was observed. All the correlations with the ECVI-38 scale were significant and negative, so when there was worse upper limb functionality, there was worse QoL.

## 4. Discussion

The aim of this study was to validate the ARAT assessment instrument for post-stroke patients in the Spanish population. Validating an evaluation instrument is necessary so that its use offers us evidence in the results and serves as a guarantee of validity and reliability. To perform a validation process, it is first carried out in the original language and then in the different versions that exist when making modifications and/or translations are validated [42].

Our results show that the Cronbach’s alpha coefficient obtained was 0.96, which indicated a very good internal consistency of the tool, since it was greater than 0.8. In addition, Cronbach’s alpha with each of the suppressed elements oscillated between 0.958 and 0.960, and the total correlation of corrected elements was greater than 0.42 in all cases, indicating that none of the items should be removed from the tool. The correlation between the items and the total score also indicated that there was a good homogeneity of the statements that composed it, and that, a priori, all the items would be measuring the same construct [43].

On the other hand, the Spearman–Brown coefficient obtained by the half-and-half test confirmed the good internal consistency of the ARAT, with a score higher than 0.80. In short, the high internal consistency demonstrated by the ARAT guarantees the homogeneity of its items, so that they measure a single construct, and it also guarantees the linear relationship between the sum of the scores of the items and the construct measured [43].

Regarding validity, it was decided to assess criterion validity, convergent validity and discriminant validity, but content validity was not evaluated, although quantitative tools that aim to collect information on the importance of a variable must verify its content validity through an analysis of the concept expressed in the variable [42]. 

The most commonly used content validation processes involve the assessment of the items of the scale by a panel of experts; however, the instrument in this study is widely used in clinical practice, which would imply that it has the support of professionals, in addition to being recommended by CAULIN [12] and SRRR [13]. In addition, it is an instrument of which the translation into Spanish has been used in other studies [17,21,23,24,44,45], so it is considered that this content validity process has already been performed for this version of the tool.

The total score of the ARAT and that of each of its subscales also correlated positively with the scores obtained in the FIM–FAM, Lawton and Brody and ECVI-38 scales, which indicates that the worse the functionality of the upper limb, the greater the limitations for the performance of activities of daily living, and the worse the QoL for the patient. This association of RAAT and ADLs has also been proven in other research, so the results indicate that ARAT had good convergent validity for the sample. In the study by Seok Nam et al. [46], the significant correlation of the ARAT scale with patient functional status and performance in ADLs is affirmed; another study confirms ARAT prediction of ADL performance in acute and subacute phases of stroke [47]. Likewise, a strong relationship has been observed between improvement in the functionality of the upper limb in daily life and QoL, indicating that focusing on the treatment of motor skills and functionality of the upper limbs is fundamental for ADLs and for the improvement of the QoL of patients after a stroke [48]. Other research also highlights the importance of and the strong relationship between the functionality and motor dexterity of the upper limb and the QoL, assuming greater autonomy in the performance of ADLs and in turn positively affecting the QoL of stroke patients [49,50,51].

This strengths and limitations should be taken into account. Regarding its strengths, first of all, we should highlight that it was a multicenter study that addresses a population with specific needs and characteristics that requires validated evaluation instruments. The results yielded significant data enabling the recommendation of the use of ARAT to evaluate the functionality of the upper limb in post-stroke patients. Secondly, we should highlight that a blinded procedure was performed, since the people in charge of collecting the data were different from the people in charge of the statistical analysis.

However, it is also necessary to take into account the limitations of this research along with its results, including the fact that the research was conducted with a convenience sample. In addition, it is important to take into consideration that the value of Cronbach’s α obtained in the analysis (α = 0.96) was greater than .90, and could be indicative of redundancy or duplication [42], which would imply that one or more items were measuring exactly the same aspect of the construct; for this reason, it is necessary to expand the sample of this study so that its factorial structure can be analyzed, although there are studies that have also addressed this aspect [52,53].

## 5. Conclusions

The Spanish version of the ARAT is a tool that demonstrates good validity and reliability for measuring upper limb functionality in post-stroke patients.

The internal consistency of the ARAT demonstrates good reliability. The ARAT also demonstrates good criterion validity when compared to the gold standard, and furthermore, its convergent validity is also appropriate.

It is recommended, therefore, that the ARAT tool be used as part of the initial assessment of a patient with stroke before they receive rehabilitation treatment, with the aim of designing an individualized plan adapted to the needs of the patient. In addition, its use is recommended for monitoring and checking the results of the treatment. 

## Figures and Tables

**Table 1 ijerph-19-14918-t001:** Half-and-half method reliability statistics.

Cronbach’s alpha	Part 1	Value	0.924
	Part 2	N of elements	19 ^a^
		Value	0.911
		N of elements	19 ^b^
	Total N of elements		38
Correlation between forms			0.941
Spearman–Brown coefficient	Equal length		0.969
	Uneven length		0.969
Coefficient of two Guttman halves			0.969

^a^ The elements were: cube 10 cm left, cube 2.5 cm left, cube 5 cm left, cube 7.5 cm left, tennis ball left, left stone, cube 10 cm right, cube 2.5 cm right, cube 5 cm right, cube 7.5 cm right, right tennis ball, right stone, pour water from one glass to another left, tube 2.5 cm left, tube 1 cm left, golilla on a left peg, pour water from one glass to another right, tube 2.5 cm right, tube 1 cm right. ^b^ The elements were: golilla on a right peg, ball 6 mm thumb-ring left, ball 1.5 cm thumb-index left, ball 6 mm thumb-heart left, ball 6 mm thumb-index left, marble thumb-ring left, marble thumb-heart left, ball 6 mm thumb-ring right, ball 1.5 cm thumb-index right, ball 6 mm thumb-heart right, ball 6 mm thumb-index right, ball 6 mm thumb-index right, marble thumb-ring right, marble thumb-ring right, right thumb-heart marble, hand behind left head, hand on left head, hand to left mouth, hand behind right head, hand on right head, hand to right mouth.

**Table 2 ijerph-19-14918-t002:** Pearson ARAT—Fulg–Meyer correlation.

		Motor Function FMA-UE	Total FMA-EU
Thick grip (left)	Correlation coefficient	0.493 **	0.439 **
	Sig. (bilateral)	<0.001	<0.001
	N	83	83
Thick grip (right)	Correlation coefficient	0.596 **	0.595 **
	Sig. (bilateral)	<0.001	<0.001
	N	83	83
Grip (left)	Correlation coefficient	0.509 **	0.454 **
	Sig. (bilateral)	<0.001	<0.001
	N	83	83
Grip (right)	Correlation coefficient	0.584 **	0.585 **
	Sig. (bilateral)	<0.001	<0.001
	N	83	83
Clamp (left)	Correlation coefficient	0.369 *	0.343 **
	Sig. (bilateral)	0.001	0.001
	N	83	83
Clamp (right)	Correlation coefficient	0.540 **	0.554 **
	Sig. (bilateral)	<0.001	<0.001
	N	83	83
Thick Mov (left)	Correlation coefficient	0.496 **	0.455 **
	Sig. (bilateral)	<0.001	<0.001
	N	83	83
Mov thick (right)	Correlation coefficient	0.588 **	0.609 **
	Sig. (bilateral)	<0.001	<0.001
	N	83	83
Total ARAT (left)	Correlation coefficient	0.469 **	0.426 **
	Sig. (bilateral)	<0.001	<0.001
	N	83	83
Total ARAT (right)	Correlation coefficient	0.580 **	0.589 **
	Sig. (bilateral)	<0.001	<0.001
	N	83	83

FMA-EU: Fugl–Meyer assessment upper extremity; ARAT: action research arm test; Left: left; Right: right. ** Correlation is significant at level 0.01 (bilateral). * The correlation is significant at level 0.05 (bilateral).

**Table 3 ijerph-19-14918-t003:** Pearson correlation ARAT—FIM–FAM, Lawton and Brody and ECVI-38.

		FIM–FAM	Lawton and Brody	ECVI-38
Thick grip (left)	Correlation coefficient	0.278 *	0.244 *	−0.307 **
	Sig. (bilateral)	0.011	0.026	0.005
	N	83	83	83
Thick grip (right)	Correlation coefficient	0.507 **	0.293 **	−0.469 **
	Sig. (bilateral)	<0.001	0.007	<0.001
	N	83	83	83
Grip (left)	Correlation coefficient	0.266 *	0.238 *	−0.284 **
	Sig. (bilateral)	0.015	0.030	0.009
	N	83	83	83
Grip (right)	Correlation coefficient	0.478 **	0.259 *	−0.445 **
	Sig. (bilateral)	<0.001	0.018	<0.001
	N	83	83	83
Clamp (left)	Correlation coefficient	0.853 **	0.168	−0.516 **
	Sig. (bilateral)	<0.001	0.129	<0.001
	N	83	83	83
Clamp (right)	Correlation coefficient	0.172	0.277 *	−0.230 *
	Sig. (bilateral)	0.121	0.011	0.037
	N	83	83	83
Thick Mov (left)	Correlation coefficient	0.485 **	0.255 *	−0.469 **
	Sig. (bilateral)	<0.001	0.020	<0.001
	N	83	83	83
Mov thick (right)	Correlation coefficient	0.505 **	0.251 *	−0.447 **
	Sig. (bilateral)	<0.001	0.022	<0.001
	N	83	83	83
Total ARAT (left)	Correlation coefficient	0.237 *	0.214	−0.283 **
	Sig. (bilateral)	0.031	0.053	0.010
	N	83	83	83
Total ARAT (right)	Correlation coefficient	0.506 **	0.283 **	−0.469 **
	Sig. (bilateral)	<0.001	0.010	<0.001
	N	83	83	83

FIM–FAM: Functional independence measure–functional assessment measure; ECVI-38: Stroke Quality of Life Scale; ARAT: Action Research Arm Test; Left: left; Right: right. ** Correlation is significant at level 0.01 (bilateral). * The correlation is significant at level 0.05 (bilateral).

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
