# Peer review of "Psychometric Properties of the Action Research Arm Test (ARAT) Scale in Post-Stroke Patients—Spanish Population"

_ijerph, 2022, doi:10.3390/ijerph192214918_

Round 1
Reviewer 1 Report
Psychometric properties of the Action Research Arm Test (ARAT) scale in post-stroke patients. Spanish Population
I appreciate the opportunity to evaluate the manuscript #ijerph-1995416 “Psychometric properties of the Action Research Arm Test (ARAT) scale in post-stroke patients. Spanish Population”.
Abstract: “The validation of measuring instruments in the field of health is a requirement to be able to use them safely and reliably. The Action Research Arm Test (ARAT) tool is an instrument validated in numerous countries and languages and for different populations, which use is widespread. The objective of this research was to determine the psychometric properties of ARAT for a sample composed of post-stroke patients. To do this, internal consistency tests were performed using Cronbach's alpha, Correlations between items and item-total and half-level test to verify their reliability. Regarding validity, criteria validity tests were performed taking the motor function dimension of the Fulg-Meyer scale as Gold Standard; and convergent validity tests were performed by correlation with the FIM-FAM, ECVI-38 and Lawton and Brody scales. The results show a very good internal consistency; as well as good criterion and convergent validity. In conclusion, the ARAT can be considered a valid and reliable instrument for the evaluation of upper limb function in post-stroke patients.”
It was not possible to perceive any theoretical basis for carrying out the investigation, however, as it is an extremely relevant object of study for public health, the absence of this contribution does not diminish the relevance of the findings.
It is a study with citation potential, with a limited methodology whose results are significant for public health. The Strengthening the Reporting of Observational Studies in Epidemiology (STROBE) protocol was used to evaluate this manuscript.
In the introduction, the authors present an important overview of the consequences of stroke cases and point to the Action Research Arm Test (ARAT) as an important scale to assess the functionality of the upper limbs of affected patients. The introduction ends with an explanation of the importance of validating this instrument for the Hispanic population.
The method is delimited, but does not meet all STROBE items, requiring a review of the following indications:
1. Indicate the study design in the title or abstract, with a commonly used term;
2. Detail the theoretical framework and reasons for carrying out the research;
3. Describe the relevant context, locations and dates, including periods of recruitment, exposure, follow-up and data collection;
4. Clearly define all outcomes, exposures, predictors, potential confounders, and effect modifiers;
5. Specify all measures taken to avoid potential sources of bias;
6. Explain how the sample size was determined;
The review of the research protocol by a Research Ethics Committee was mentioned, thus respecting the international provisions in force.
The presentation of the results is clear and the psychometric calculations seem to be rigorously tested, thus ensuring their validation.
The discussion presents other studies that used the instrument translated into Spanish and the data corroborate those presented by the authors.
I also emphasize the fact that 50% of the references used were published more than 5 years ago, requiring an update of the study sources.
Aware of the quality of this research, I indicate my assent to the publication after the modifications of the aforementioned weaknesses.
Author Response
Mrs. Jessica Fernández Solana
Department of health sciences
University of Burgos, Paseo Comendadores s/n.
Burgos, 09001, Spain
Tel. (+34) 947499108
Email: jfsolana@ubu.es
31-10-2022
IJERPH. Subject: Submissions Needing Revision
Dear editor.
Thank you very much for inviting us to submit our response to reviewers for our manuscript (ijerph-1995416) entitled: “Psychometric properties of the Action Research Arm Test (ARAT) scale in post-stroke patients. Spanish Population.”
We have checked our manuscript according to the Academic Editor, the reviewers’ comments and the Journal requirements. We have also responded to some comments from reviewers point by point).
We would be very grateful if you could consider our manuscript to be published in your journal.
Yours sincerely,
Jessica Fernández Solana, OT, PT
- Response to Reviewer 1:
First of all, we would like to express our sincere gratitude for all comments and suggestions received from the Reviewer 1. This information has certainly enriched the text for its best understanding, thank you very much indeed. We have clarified the reviewer1’s questions. We have introduced the required changes both in our answers to the specific comments and in the final manuscript V2.
Psychometric properties of the Action Research Arm Test (ARAT) scale in post-stroke patients. Spanish Population
I appreciate the opportunity to evaluate the manuscript #ijerph-1995416 “Psychometric properties of the Action Research Arm Test (ARAT) scale in post-stroke patients. Spanish Population”.
It was not possible to perceive any theoretical basis for carrying out the investigation, however, as it is an extremely relevant object of study for public health, the absence of this contribution does not diminish the relevance of the findings.
It is a study with citation potential, with a limited methodology whose results are significant for public health. The Strengthening the Reporting of Observational Studies in Epidemiology (STROBE) protocol was used to evaluate this manuscript.
In the introduction, the authors present an important overview of the consequences of stroke cases and point to the Action Research Arm Test (ARAT) as an important scale to assess the functionality of the upper limbs of affected patients. The introduction ends with an explanation of the importance of validating this instrument for the Hispanic population.
The method is delimited, but does not meet all STROBE items, requiring a review of the following indications:
- Indicate the study design in the title or abstract, with a commonly used term;
- Detail the theoretical framework and reasons for carrying out the research;
- Describe the relevant context, locations and dates, including periods of recruitment, exposure, follow-up and data collection;
- Clearly define all outcomes, exposures, predictors, potential confounders, and effect modifiers;
- Specify all measures taken to avoid potential sources of bias;
- Explain how the sample size was determined;
Response: Thank you for your comments.
We have indicated the study design consists on a “psychometric analysis” in the abstract.
We have made some changes in the introduction to detail the reasons for conducting this study, based on the literature (See lines 76-103)
We have described the context (See lines 110-115)
We have modified instruments section in order to follow your recomendations (See lines 118-122)
Although we had already included that it consisted of a convenience sample in the limitations of the study we have also specified that no sample calculation was made in the procedure section.
The review of the research protocol by a Research Ethics Committee was mentioned, thus respecting the international provisions in force.
The presentation of the results is clear and the psychometric calculations seem to be rigorously tested, thus ensuring their validation.
The discussion presents other studies that used the instrument translated into Spanish and the data corroborate those presented by the authors.
I also emphasize the fact that 50% of the references used were published more than 5 years ago, requiring an update of the study sources.
Response: Thank you for your comments. We have updated the references to include some more recent ones. Some of the older references, in many cases, are publications that explain the contextualisation, development and validation procedures of the ARAT tool in other countries and populations.
Aware of the quality of this research, I indicate my assent to the publication after the modifications of the aforementioned weaknesses.
We hope we have now answered all your comments and we are looking forward to hearing from you again.
Thank you very much,
Jessica Fernández Solana, OT, PT

Reviewer 2 Report
I would like to thank you for the opportunity given to me to review this manuscript.
The introduction is very fluid and easy to understand. I congratulate the authors because it has been a long time since I read such a well-written introduction.
I consider that the instrument section is very well detailed.
Were there exclusion criteria?
The people who carried out the investigation, were they blinded? This is important as it can condition the measurement. If not, it should be named as limitations of the study.
The results, discussion and conclusions section is clear and concise.
Author Response
Mrs. Jessica Fernández Solana
Department of health sciences
University of Burgos, Paseo Comendadores s/n.
Burgos, 09001, Spain
Tel. (+34) 947499108
Email: jfsolana@ubu.es
07-11-2022
IJERPH. Subject: Submissions Needing Revision
Dear editor.
Thank you very much for inviting us to submit our response to reviewers for our manuscript (ijerph-1995416) entitled: “Psychometric properties of the Action Research Arm Test (ARAT) scale in post-stroke patients. Spanish Population.”
We have checked our manuscript according to the Academic Editor, the reviewers’ comments and the Journal requirements. We have also responded to some comments from reviewers point by point).
We would be very grateful if you could consider our manuscript to be published in your journal.
Yours sincerely,
Jessica Fernández Solana, OT, PT
- Response to Reviewer 2:
First of all, we would like to express our sincere gratitude for all comments and suggestions received from the Reviewer 2. This information has certainly enriched the text for its best understanding, thank you very much indeed. We have clarified the reviewer2’s questions. We have introduced the required changes both in our answers to the specific comments and in the final manuscript V2.
I would like to thank you for the opportunity given to me to review this manuscript.
The introduction is very fluid and easy to understand. I congratulate the authors because it has been a long time since I read such a well-written introduction.
I consider that the instrument section is very well detailed.
Were there exclusion criteria?
The people who carried out the investigation, were they blinded? This is important as it can condition the measurement. If not, it should be named as limitations of the study.
The results, discussion and conclusions section is clear and concise.
Response: Thank you very much for your comments.
We have added exclusion criteria (see line 174-175)
A randomized, controlled and blinded sampling was performed (See line 169-171 and lines 306-309)
We hope we have now answered all your comments and we are looking forward to hearing from you again.
Thank you very much,
Jessica Fernández Solana, OT, PT
